# Packing Incubation and Addition of Rot Fungi Extracts Improve BTEX Elimination from Air in Biotrickling Filters

**DOI:** 10.3390/molecules29184431

**Published:** 2024-09-18

**Authors:** Piotr Rybarczyk, Krzysztof Cichon, Karolina Kucharska, Dominik Dobrzyniewski, Bartosz Szulczyński, Jacek Gębicki

**Affiliations:** 1Department of Process Engineering and Chemical Technology, Faculty of Chemistry, Gdańsk University of Technology, Narutowicza 11/12 Street, 80-233 Gdańsk, Poland; karolina.kucharska@pg.edu.pl (K.K.); dominik.dobrzyniewski@pg.edu.pl (D.D.); bartosz.szulczynski@pg.edu.pl (B.S.); jacek.gebicki@pg.edu.pl (J.G.); 2Waste Utilization Facility Ltd. in Gdańsk, Jabłoniowa 55 Street, 80-180 Gdańsk, Poland; kcichon@zut.com.pl; 3Implementation Doctoral School, Gdańsk University of Technology, Narutowicza 11/12 Street, 80-233 Gdańsk, Poland

**Keywords:** biotrickling filtration, air treatment, BTEX, fungi extracts, inoculation pattern, process performance

## Abstract

The removal of benzene, toluene, ethylbenzene, and xylene (BTEX) from air was investigated in two similar biotrickling filters (BTFs) packed with polyurethane (PU) foam, differing in terms of inoculation procedure (BTF A was packed with pre-incubated PU discs, and BTF B was inoculated via the continuous recirculation of a liquid inoculum). The effects of white rot fungi enzyme extract addition and system responses to variable VOC loading, liquid trickling patterns, and pH were studied. Positive effects of both packing incubation and enzyme addition on biotrickling filtration performance were identified. BFF A exhibited a shorter start-up period (approximately 20 days) and lower pressure drop (75 ± 6 mm H_2_O) than BTF B (30 days; 86 ± 5 mm H_2_O), indicating the superior effects of packing incubation over inoculum circulation during the biotrickling filter start-up. The novel approach of using white rot fungi extracts resulted in fast system recovery and enhanced process performance after the BTF acidification episode. Average BTEX elimination capacities of 28.8 ± 0.4 g/(m^3^ h) and 23.1 ± 0.4 g/(m^3^ h) were reached for BTF A and BTF B, respectively. This study presents new strategies for controlling and improving the abatement of BTEX in biotrickling filters.

## 1. Introduction

Air pollution containing volatile organic compounds (VOCs) is a serious problem for both human health and the environment. VOCs can be removed from the air using different methods, including absorption, adsorption, incineration, low-temperature plasma, photocatalytic oxidation, and biological methods [1,2]. Out of the mentioned methods, biological ones have several advantages that are highly appreciated currently, including low energy consumption, safety, the generation of no or little secondary waste, sustainability, and eco-friendliness. This is why biological air purification, i.e., biofiltration, biotrickling filtration, and bioscrubber technology, are considered to be green technologies and are attracting more and more attention from both industry and researchers [3]. 

The development of biofiltration technology for air treatment started with the establishment of conventional biofilters. Such bioreactors utilize VOCs as carbon and energy sources for microorganisms, inhabiting the biofilter packing, which is made of natural/biological materials (e.g., peat, barks, wood chips) [4,5]. As a result of microbial metabolic activity, VOCs are converted into carbon dioxide, biomass, and water. Due to several problems encountered in conventional biofiltration technology, such as packing compaction and degradation, leading to increased resistance to gas flow and limited possibilities of process condition management, biotrickling filters (BTFs) have been developed [3]. BTFs offer complex process control and regulation possibilities, allowing for the adjustment of liquid-phase pH, composition, or watering frequency as well as longer operation times when compared to conventional biofilters [6,7]. Similar to conventional biofilters, in BTFs, polluted air passes through a porous layer. The BTF packing is populated with microorganisms capable of biologically decomposing pollutants. Considering the net effect of adsorption, absorption, and diffusion phenomena, the contaminants become available for biodegradation by microorganisms, forming a biofilm that covers the surface of the packing elements. A trickling liquid solution of mineral salts is sprayed over the packing [8,9]. After the polluted gas passes through the biotrickling filter, the purified gas leaves the biofilter from the top.

In this work, benzene, toluene, ethylbenzene, and xylene were chosen as target air pollutants. These VOCs, typically abbreviated as BTEX, are associated with adverse effects on human health, including damage to the liver, nervous system, heart, and kidneys. From a chemical viewpoint, BTEX represents compounds of low and medium solubility in water, and according to the values of Henry’s law, constants are categorized as highly or medially hydrophobic [10]. The investigated VOCs are found in waste gasses emitted from municipal waste rendering plants and various branches of chemical and related industries. The selected physicochemical parameters of BTEX are given in Table 1.

The removal of ethylbenzene, styrene, xylene, and toluene from air using biotrickling filters has been studied for several years (Table 2). However, the majority of available results refer to studies in which single VOCs were treated in biotrickling filters. When mixtures of VOCs undergo biotrickling filtration, interactions between the mixture components affect the course and performance of the biofiltration process [11,12,13,14]. These interactions may result in neutral, positive, or negative effects on the process efficiency due to changes in the physicochemical and biological aspects of the process. Thus, studies focusing on VOC mixtures are important and close to real-world and industrial concerns.

**Table 1 molecules-29-04431-t001:** Selected physicochemical parameters of investigated mixture of volatile organic compounds *****.

Compound	Solubility in Water, mg dm^−3^ (25 °C)	Henry’s law Constant, mol m^−3^ Pa^−1^ [15] **	P_o/w_ (20 °C)	Boiling Point, °C (101,325 Pa)
Benzene	1770	0.0014–0.0018	2.13	90.1
Toluene	580	0.0013–0.0017	2.73	110.0
Ethylbenzene	200	0.0011–0.0013	3.60	136.1
Xylene (mixture of xylenes)	156	0.0012–0.0023	3.15	139.1

* Data taken from safety data sheets available from respected chemicals manufacturers. ** Ranges for the most frequently reported values are listed.

**Table 2 molecules-29-04431-t002:** Selected research on the removal of toluene, ethylbenzene, xylene, styrene and BTEX mixture from waste gases in biotrickling filters.

Compound/s	Packing Material/Volume of Packing	Inoculum	Inlet Loading, g m^−3^ h^−1^	Removal Efficiency, %	EBRT, s	Trickling Liquid Pattern	Reference
Benzene	Zeolite-contained polyethylene media/7.5 dm^3^	Defined microorganism consortium with *Bacillus cereus 1*	100	70	40	Continuous, 4 m^3^ per 1 m^3^ of packing	[16]
	Polypropylene spheres and fibre balls/16 dm^3^	No information	67	65	30	Intermittent, 3.5 dm^3^ of mineral medium once per hour	[17]
	Pelletized diatomaceous earth (Celite)/2.7 dm^3^	Activated sludge/packing with developed microorganisms from other biofilter	48	90	120	2 dm^3^ per day (buffered mineral medium)	[18]
Toluene	Ceramsite/ 4 dm^3^	*Burkholderia sp.* strain T3 isolated from WWTP activated sludge	474	98	32	Intermittent spraying, 0.2 mL/s	[19]
	Polyurethane foam cubes/ 5 dm^3^	Acclimated activated sludge	600	70	30	sulphate-free mineral salt medium (MSM); 0.00048 m^3^/h	[20]
	Slags/144 dm^3^	Dried yeast powder	55	85	76	Spraying 32 m/h	[21]
	Pall rings and pumice/2 dm^3^	*Ralstonia eutropha*	200	85	45	Spraying 20 mL/min	[22]
	Glass beads/ 3.9 dm^3^	Inoculum taken from previously working BTF	360	97	28	Differential biotrickling filter	[23]
	Ceramic pellets/ 4.27 cm^3^	Fungi BTF: *Fusarium*, *Paramicrosporidium saccamoebae* (from activated sludge and cultivation fungi-oriented)	70	82	77	Intermittent trickling pattern (inorganic mineral medium)	[24]
Ethylbenzene	Open-pore reticulated polyurethane sponge	Activated sludge from WWTP	189	69	40	Spraying for 3 s every 3 min; 4.5 L/day, modification of trickling liquid with surfactant and Zn(II)	[25]
	Polyurethane sponge	Fresh biological sludge from WWTP	264	50	30	0.2 L/h; Biosurfactant addition to liquid phase	[26]
	Polyurethane sponge	Activated sludge from WWTP	180	80	30	4.8 L/day; Addition of saponins to liquid phase	[27]
Xylene (isomers)	Ceramic particles/4.7 dm^3^	*Bacillus firmus*	1450	98	84.8	Continuous trickling	[28]
	Packing material with porosity of 0.95/1.7 dm^3^	Enriched mixed culture from activated sludge of pharmaceutical plant	80	87.5	90	Silicone-oil addition (5% *v*/*v*), continuous trickling	[29]
	Diatomaceous earth pellets/ 2.4 dm^3^	Enriched activated sludge from wastewater treatment plant	150	73	25	Intermittent spraying, 100 mL once each 3 h	[30]
BTEX	Waste blue mussel shells/6 m^3^	Effluent from refinery wastewater treatment plant	2	76	60	Continuous trickling, 0.9 m^3^ h^−1^	[31]
	Polyurethane foam/1 dm^3^	Microbial consortium enriched from petroleum polluted soil	100	60	30	Continuous trickling with M9 medium and vitamins	[32]
	Kaldnes rings/ 2 dm^3^	Activated sludge from denitrification-nitrification wastewater treatment plant	5.7	30–60	1800	Mineral salt medium, 2 m h^−1^	[33]

Bacterial and fungal co-cultures are successfully applied in the remediation and disposal of waste gases and odors [34,35,36]. Such co-cultures eliminate or decompose harmful substances into less toxic compounds. The biodegradation of pollutants can be enhanced by adjusting the parameters of the primary influence on microbial growth, thus increasing not only their number but also the degradation activity. Cooperation between fungi and bacteria is advantageous [37,38] since different oxygen routes are adopted. As a result, different products of VOC degradation are formed, i.e., carbon dioxide, methane, sulfur, and nitrogen oxides or phenol derivatives.

In this work, the addition of lignocellulose-originating white rot fungi enzymes to a trickling liquid was investigated to verify the potential of such a liquid-phase modification to enhance the removal of hydrophobic VOCs from air in the biotrickling filtration process. Depending on the species and the plant origin, the amount and composition of lignin vary. Lignin monomers contain aromatic alcohols: p-coumaryl, coniferyl, and sinapine [39]. Their distinguishing feature is the number of places substituted with a methoxyl group. The biopolymer is created as a result of a series of reactions, mainly according to the radical polymerization mechanism between lignin monomers, i.e., p-hydroxyphenyl unit, p-coumaryl alcohol, the guaiacol, and coniferyl alcohol [40]. Lignin plays a supporting role in plants, mainly providing mechanical and antimicrobial strength. For this reason, effective methods of purifying lignin, isolating it from plant organisms, and obtaining valuable products are given attention. Mechanical methods are not efficient; therefore, ligninolytic enzymes are used. White rot fungi produce a non-specific system characterized by a high oxidation reduction potential, thanks to which these microorganisms are used to degrade many xenobiotics. So far, several species of basidiomycetes white rot fungi have been characterized as capable of the complete mineralization of polychlorophenols, including *Phanerochaete chrysosporium, Trametes versicolor, Panus tigrinus, Pleurotus pulmonarius* [41]. Fungi degrade lignin by secreting lignynases, i.e., phenol oxidases (laccase), and heme peroxidases. Laccases use oxygen as electron acceptors, and peroxidases require a co-substrate, i.e., hydrogen peroxide. Depending on the species and environment, white rot fungi variously secrete several lignin-modifying enzymes for effective lignin degradation [42]. White rot fungi enzymes can support the functioning of the microbial biofilm during biofiltration due to the non-specific nature of lignolytic enzymes, which are also capable of decomposing organic substances, including VOCs. White rot fungi and the lignin-modifying enzymes they produce are capable of degrading a wide range of trace organic pollutants but are suspected of adversely affecting human health and the condition of fauna and flora [43]. Currently, white rot fungi are used to remove compounds, such as pharmaceuticals, polycyclic aromatic hydrocarbons, polychlorinated biphenyls, chlorinated solvents, and even textile dyes [44]. Overall, compounds with strong electron-donating groups, such as amine, hydroxyl, alkoxy, alkyl, and acyl, are expected to be effectively eliminated. In contrast, compounds containing strong electron-withdrawing groups are difficult to remove (amide groups, carboxyl groups, halogen groups, and nitro groups). Therefore, lignolytic enzymes are believed to enhance air purification from BTEX using biotrickling filtration [45,46,47].

The aim of this work was twofold: one was the systematic comparison of the effects of the inoculation procedure on the BTF performance; the second was to investigate the effects of using white rot fungi enzyme extracts to enhance the abatement of mixed hydrophobic VOCs in biotrickling filters. The literature evidence shows two popular methods of packing inoculation, namely soaking the packing materials in the liquid medium containing the microbial cultures prior to placing the elements in the biofilter (e.g., as reported by Dou et al. [30]) or via circulation of the liquid medium with the inoculum through the packing elements already placed in the biofilter (e.g., as reported by Wu et al. [29], which is a popular strategy for BTF inoculation). However, a comparison of the effects of such inoculation approaches on the system performance is missing in the literature. Additionally, the use of rot fungi extracts to improve the colonization of the biotrickling filter seems to be an easy and cheap solution for expected performance enhancements and has not been tested in biotrickling filtration systems for air treatment.

The problem of improving biotrickling filtration performance for the removal of hydrophobic volatile organic compounds from the air is still challenging. It has been mainly addressed by developing the following methods: use of surface active substances, mixing of hydrophobic VOCs with hydrophilic ones, use of the two-phase system, two-stage processes, or utilization of fungi [2,48,49,50,51]. Thus, investigations on the use of rot fungi as well as evaluation of the effects of biofilter inoculation on the biofiltration performance carried out in this work are novel and present practical aspects for possible applications in the industrial systems for BTEX abatement, e.g., improving the treatment of waste gases from composting plants in municipal waste rendering facilities.

## 2. Results

### 2.1. Design of Experiment

Before starting the biotrickling filtration experiments, the effects of parameters influencing the development of the biofilm layer on the packing elements were examined. *The* selection of statistically important variables for the incubation procedure applied to prepare biofilter packing was based on experiments performed according to the Plackett–Burman statistical design (Table 3 and Table 4). Four statistical variables and one discrete variable were investigated. The type of inoculation was used as a discrete variable, assuming level −1, depicting the inoculation via liquid circulation in the biotrickling filter, without any preparation of the packing elements, and level 1 indicating biotrickling filter inoculation preceded by the incubation of packing elements. The statistical variables included the inoculation time before the introduction of VOC mixtures, the use of shaking, increased temperature, and pH adjustment from the natural value characteristic of the activated sludge suspension to a value similar to the conditions prevailing in the biofilter, i.e., pH = 5.1. The results of the applied procedure were assessed based on the biofilm layer development. The range of parameters is listed in Table 4.

Figure 1 presents a Pareto chart for the factors that have a significant influence on biofilm formation. Analysis of the results of the statistical plan allows us to conclude that the use of the incubation procedure by shaking the packing elements in the presence of microorganisms and a longer (24 h) incubation time favors the formation of a biofilm of a specific structure (Figure 1a). The correlogram (Figure 1b) shows that in the tested range of variability, a lower value of the biofilm layer (corresponding to the expected thickness of the biofilm layer) was obtained at a lower temperature, maximum mixing, and with the longest incubation time. These results allowed us to determine two sets of process conditions based on the Plackett–Burmann plan (Table 5), which allows for obtaining a diverse structure of the biofilm layer, which may affect the process performance. 

### 2.2. Performance of Biotrickling Filters and pH Variations

The performance of BTEX removal from the air was investigated in two biotrickling filters, differing in the inoculation procedure, i.e., biotrickling filter A (BTF A) with packing elements inoculated with a soaking method and biotrickling filter B (BTF B), inoculated via continuous circulation of a solution containing inoculum. The parameters of biotrickling filtration processes are given in Table 6. Process performances show the removal efficiencies of BTF A and BTF B in Figure 2. 

Changes in the removal efficiency of VOCs during the biotrickling filtration present similar variations for all investigated compounds. In Figure 2a–d, three main phases of the process can be distinguished. In phase one, the process starts up with low values of removal efficiency (<0.1), followed by a more rapid (BTF A) or smoother (BTF B) increase in the BTEX removal efficiency, and then the removal efficiency values stabilize within about 30 days from the process initiation. Then, in the second phase, around the 35th–40th day from the process start-up, the removal performance drops drastically. In the third phase, after day 55, the removal performance for all BTEX rapidly increases and stabilizes around the 70th day, keeping similar removal efficiency values until the end of investigations (120 days). The results show that removal efficiencies for all BTEX compounds are higher for BTF A than for BTF B. Additionally, the start-up period for BTF A is shorter than BTF B, accounting for about 20 and 30 days, respectively. In all cases, the BTEX removal efficiencies reached in the third phase are higher than in the first phase. The highest average removal efficiency of BTEX (RE = 0.96; the sum of BTEX), corresponding to the highest average elimination capacity (EC), was reached for BTF A in the final days of the experiment, with the average value of EC being about 28.8 ± 0.4 g/(m^3^ h) for the inlet VOC loading (IL) of about 30 ± 0.5 g/(m^3^ h). The value of the removal efficiency depends on the inlet loading and other process parameters (e.g., pH, empty bed residence time, EBRT, or trickling pattern). Interestingly, the highest RE values obtained in this research are higher than in other reported results for comparable or lower inlet loading (Table 2). The following order of increasing removal efficiency for investigated VOCs can be proposed: benzene, xylene, ethylbenzene, and toluene. For example, Torretta and co-workers obtained both a similar BTEX removal efficiency using water scrubbing and biotrickling filter, for inlet loading of about 2.1 g/(m^3^ h), and similar order of BTEX removal efficiency [52].

The pattern of variations in the biotrickling filtration performance is related to changes in the pH of the trickling liquid (Figure 3). The pH of the circulating trickling liquid was about 7 during the process initiation. During the first 20 days, the pH slightly decreases, keeping the values between 6.5 and 7, depending on the liquid makeup in the trickling solution tank. During the initial 20 days of the process, a mineral salt medium (MSM) was used. Then, tap water was used from the 21st to the 55th day of the process. The pH of the trickling liquid decreased from day 21 until day 55, dropping from about 6.5 to 4.0 and leading to BTF acidification. Then, rot fungi extract solution with fresh MSM liquid was introduced, resulting in a sharp pH increase to about 6–6.5. A pH of about 6.0 was kept until the end of investigations, using the MSM solution with a makeup period of about 7 days. The pH values were similar for both BTF A and BTF B systems.

### 2.3. Production of CO_2_

The production of biodegradation-generated CO_2_ as a function of elimination capacity for the sum of BTEX is shown in Figure 4. CO_2_ production increases linearly with an increase in the elimination capacity, and CO_2_ production is slightly higher for BTF A than BTF B. The values of CO_2_ production are provided for variable inlet loading, tested between the 90th and 110th days of the process (variations in the removal efficiency values reported in Figure 1 occur for this period). The average ratio of the carbon dioxide production to the elimination capacity was 2.71 for BTF A and 2.37 for BTF B. 

### 2.4. Pressure Drop across the Biotrickling Filter Packing

Changes in the pressure drop across the packed layers of BTF A and BTF B are shown in Figure 5. The pressure drop is an important process parameter, reflecting the performance capability and biomass development in the biotrickling filter [53]. Along with the formation and development of the biofilm during the biofiltration process, the pressure drop (ΔP) increases. This was observed for both investigated systems. Moreover, the pressure drop was slightly higher for BTF B than for BTF A. It was found that the dry mass of biomass developed in the packing of BTF A was 30 ± 0.6 g dm^−3^, while for BTF B, it was 21 ± 0.7 g dm^−3^. However, the moisture content of the single packing element for BTF A was about 77 ± 2%, and for BTF B, it was about 84 ± 3%. A higher tendency to hold the liquid in the biofilter packing/biofilm layer for BTF B may justify higher resistance to the gas flow (i.e., higher pressure drop) through the BTF B than the biofilter BTF A system. Additionally, a thinner water layer around the biofilm, i.e., lower water content per single packing element, facilitates the availability of VOCs to the biofilm, resulting in higher biofiltration performance, especially for hydrophobic VOCs [54]. This was observed for BTF A, having lower ΔP than BTF B. A higher moisture content of BTF B, resulting from continuous trickling in the first 40 days of biofiltration, may also lead to clogging of the voids by water and a higher tendency of unwanted biofilm detachment, resulting in further void clogging and increased pressure drop [55].

### 2.5. Surface Tension and Zeta Potential of Trickling Liquid 

Changes in the physicochemical parameters of trickling liquid, i.e., surface tension and zeta potential, are shown in Figure 6. Samples collected after 10, 30, 90, and 120 days of biofiltration were evaluated. Both the values of surface tension and the absolute values of the zeta potential decreased with the biofiltration time, showing the lowest values at the end of investigations. For both investigated parameters, their values (or absolute values) were higher for BTF B than for BTFA.

### 2.6. Changes in BTEX Concentrations in the Trickling Liquid

Changes in BTEX concentrations found in the headspace of the trickling liquid samples are presented in Figure 7. BTEX concentrations in the liquid phase are higher for the samples collected after a longer period of the biofiltration process. The highest values of VOC concentrations were found for toluene and the lowest for xylene. The respective values of BTEX headspace concentrations in the trickling liquid were higher in BTF A than in BTF B.

### 2.7. Biofilm Layer Development

Figure 8 shows differences in the formation and structure of the biofilm layers developed in BTF A and B. Packing elements with microorganisms A and without incubation are shown in Figure 8a. The circles marked in the figure indicate the places where biofilm development was observed. The changes in the structure are presented in Figure 8b–d. Additionally, the macroscopic changes in the packings of BTF A and BTF B, due to the biofilm formation during the process duration, are shown in Figure A1 in the Appendix A. 

## 3. Discussion

The current research and industrial issues regarding the biological filtration of waste air aim at improving the abatement of hydrophobic VOCs. Improvement strategies mainly involve either modification of the microbial composition, allowing for enhanced biodegradation rates or applying additives to the liquid phase, resulting in the decreased mass transfer of VOCs from the gas to liquid phase. From a mathematical viewpoint, these strategies aim at affecting *D_AL_, δ_film_ or H_i_* in the following formula [56]:*R = (D_AL_/δ_film_) · a · ((C_gi_/H_i_) − C_li_)*(1)
where *R* is the overall mass transfer rate (g/(m^3^ s)), *D_AL_* is the molecular diffusivity of the pollutant in the liquid (m^2^/s), *δ_film_* is the liquid film/biofilm thickness over the packing element [m], *a* is the gas–liquid specific interfacial area [m^2^/m^3^], *C_gi_* and *C_li_* are the “*i*” pollutant concentrations in the gas and liquid phases, respectively (g/m^3^), and *H_i_* is the Henry’s coefficient for the “*i*” component (dimensionless). 

In this manuscript, the authors investigated the influence of the inoculation procedure on the efficiency of BTEX abatement in biotrickling filters. It was assumed that the inoculation procedure affects the biofilm formation, its thickness, and structure. It is expected that by developing a thinner biofilm layer, the barrier for mass transfer of VOC compounds from the gas to liquid phase can be decreased, leading to increased bioavailability of VOCs for biodegradation. Additionally, adding the white rot fungi extracts to the BTF systems increases the biotrickling filtration performance. 

The biofiltration systems were operated based on microorganisms originating from activated sludge from a wastewater treatment plant. Activated sludge is a complex biological system, in which physical processes and biochemical reactions occur simultaneously, leading to wastewater treatment. The group of organisms found in activated sludge consist of many specialized groups with precisely defined, interconnected functions. 

Different approaches towards immobilizing microbial species on the packing elements of biotrickling filters are practiced. A three-step immobilization method was adopted by Liu et al. [57]. Typically, the efficient inoculation period leading to biofilm formation, realized by direct inoculation from activated sludge, takes about 15 days, which is rather a long time in terms of possible industrial applications. The three-step immobilization consisted of subjecting the inoculum to aeration flasks with different concentrations of nutrients, lower and higher in the first and second stages, and then transferring to the biofilter with the use of lower-nutrient-content spray liquid. This reduced the inoculation period to 10 days. Marycz investigated the inoculation of polyurethane discs with fungi, and co-workers proposed a procedure lasting for 5 days to reach the complete colonization of a PU disc [58]. Additionally, it is reported that a shorter immobilization time is more valid for pure cultures than in the case of an activated sludge inoculum [57]. A two-step inoculation method was described by Zhang et al. [59]. This procedure included wetting the ceramsite particles in the activated fungal liquid with hand mixing. Then, the wetted packing elements were placed in the biofilter and supplied with toluene and nutrient solution. However, a detailed procedure is not provided. Hong et al. proposed a three-day intensive inoculation of polypropylene balls in activated sludge and glucose reach medium with subsequent transfer of the balls to the biotrickling filter [60]. Then, the simultaneous biofilm formation and acclimation to the treated VOC mixture occurred, resulting in an effective start-up period of 10 days. 

The penetration depth of the gas mixture in the biofilm is crucial in determining the conversion rate in the biofilter. If the biofilm is completely penetrated by the substrates, the maximum rate of the biochemical reaction is only reaction limited. Unfortunately, the most common situation involves the partial penetration of at least one of the substrates in a thick biofilm layer caused by diffusion resistance in the biofilm. Then, only the fine outer biofilm layer will be active concerning the degradation reaction. Therefore, in this work, the preparation of the thinnest possible biofilm on a packing element, enabling obtaining the most prominent and largest possible interfacial surface, will enable the effective use of the capabilities of bacterial cultures, shifting the balance control of the degradation of undesirable substances from the diffusive to kinetic region. The performance results of the BTF A system (Figure 2) indicate that the start-up period is shorter than in the case of BTF B and takes about 10 days (BTF A), similar to the results obtained by Hong et al. [60].

The elimination of BTEX from the air (actual waste gas stream from a refinery) was investigated by Raboni and co-workers using a missed gas treatment system, including a bioscrubber, biotrickling filter, and conventional biofilter [31]. They concluded that the effective removal of BTEX from waste gas was possible due to the complementary and synergistic performance of the bacterial and fungal consortia. In their system, a bacterial biotrickling filter allowed for the effective degradation of toluene and xylene. Meanwhile, benzene and ethyl benzene were removed in the biofilter, containing a granular peat-packing medium with fungi. Liao et al. [32] investigated the interactions between benzene, toluene, ethylbenzene, and xylene, showing that benzene removal is promoted in the presence of other VOCs from the BTEX group. The removal of hydrophobic VOCs proceeds with lower efficiency than for hydrophilic VOCs because the mass transfer limitations increase with an increase in the VOC hydrophobicity, thus leading to a lower driving force for the mass transfer from the gas to liquid phase [61,62]. According to the classification of the hydrophilic/hydrophobic character of VOCs [10,61], hydrophilic, moderately hydrophilic/hydrophobic, and hydrophobic VOCs can be distinguished based on Henry’s constant value. BTEX belongs to the group of VOCs with a moderately hydrophobic character, yet due to chemical structure differences, their bioavailability is different, and, thus, the removal performance in biotrickling filters can be different [63]. Additionally, the presence of multiple VOCs with different hydrophobicities in the treated gas mixture affects their removal performance [64]. The results reported in this study suggest the following order for the efficiency of BTEX removal from air, irrespective of the system investigated (either BTF A or BTF B). The highest removal performance occurs for toluene, then for ethylbenzene and xylene, and the lowest removal occurs for benzene. This is in accordance with other available research results [32].

When the biotrickling filter was operated with pre-incubated polyurethane foam discs (BTF A), greater efficiency in the BTEX removal was observed. However, in both tested systems, the BTEX removal efficiency decreased after approximately 40–50 days. Parallel to the decrease in efficiency, a decrease in pH was observed (Figure 2 and Figure 3), which may indicate changes in the microbial culture existing in the biofilm structure. The pH drop was mainly caused by the use of tap water, without the buffering capacity compared to the MSM solution. Testing of the use of tap water had two features to be elucidated: firstly, is it possible to use cheap and available tap water to sustain the effective biotrickling filtration of BTEX and, secondly, to investigate the system response variable pH and check the system recovery after introducing MSM. Tap water utilization as a trickling liquid proved impossible, while the application of MSM with white rot fungi extract proved to be an efficient strategy for improving BTEX abatement in biotrickling filtration systems. 

From the exploitation viewpoint, especially in industrial applications, the pressure drop is an important operational parameter in the biotrickling filtration process. This process parameter must be carefully monitored and managed. Recently, Qin et al. proposed the use of commutation to decrease the pressure drop while simultaneously retaining the process performance [65]. Pressure drop differences between BTF A and B are small; however, the ΔP value is always higher for BTF B than BTF A (Figure 4). This may be related to the trickling pattern, which was continuous during the first 40 days of the process, for BTF B. Additionally, the measurements of the mass of dry biomass and moisture content of the packings (reported in Section 2.3) indicate that, in addition to the lower biomass growth in BTF B, its water-holding capacity is higher than in BTF A. This may add to increased flow resistance for the gas phase, as well as lower performance for BTEX removal, due to the higher gas-to-liquid/biofilm mass transfer barrier. 

Carbon dioxide production is an important parameter evaluating the biotrickling filter performance. It characterizes the mineralization rate of organic compounds, indicating that the more CO_2_ is generated in the system, the less biomass is produced, thus limiting the clogging phenomenon in the biofilter packing [66]. For example, the theoretical relationship between CO_2_ production and EC for xylene is 3.32 for its complete conversion to CO_2_ and water. The proportion between CO_2_ production and elimination capacity, depicted by (M_CO2/EC_), close to the theoretical one, indicates that the mineralization rate of VOCs by the microbes is high, indicating the complete degradation of VOCs. Carbon dioxide production can decrease with an increase in VOC inlet loading, e.g., due to the inhibition of microorganisms. This feature was noted by Ramirez and co-workers in the case of ethanol biotrickling filtration [67]. Zhu and co-workers [68] investigated the biotrickling filtration of toluene and found a relationship between the toluene elimination capacity and carbon dioxide production in the process. The mass ratio of carbon dioxide production to elimination capacity (M_CO2/EC_) informs us about the mineralization degree of the VOC in the biofiltration process. The theoretical value of (M_CO2/EC_) for toluene biodegradation, assuming its complete oxidation to CO_2_ and H_2_O, equals 3.35. However, Zhu et al. received a (M_CO2/EC_) value of 1.45. The discrepancy between the theoretical and experimental (M_CO2/EC_) value can be explained by the specific biodegradation pathway of VOCs, leading to the generation of intermediate products that may not degrade immediately, biomass production as well as the accumulation of CO_2_ in the liquid phase (e.g., partially indicated by the drop in the pH value of the trickling liquid). Gallastegui et al. [69] obtained (M_CO2/EC_) values for ethylbenzene and toluene of 1.36 and 2.84, respectively. The authors attributed the rate of CO_2_ production to the microbial composition of the biofilm, depicting hither CO_2_ production of the bacterial systems and lower CO_2_ production for fungal biosystems since fungi have higher microbial growth than bacteria. Thus, in fungal systems, more carbon is used for biomass formation. Table 7 presents selected (M_CO2/EC_) values based on the literature evidence on VOC removal from air using biotrickling filters. The value of 2.71 obtained for system A in this study reveals that the pre-incubation procedure and application of rot fungi allow for high mineralization of the treated VOCs, and this approach may be recommended for use in biotrickling filters treating hydrophobic air contaminants. 

The study adopted Plactkett–Burman’s matrix for five variables, using the number of experiments typical for repeated systems, i.e., twelve repetitions, taking into account four confidence levels (Table 5). The main influence of the tested factors was determined based on the microscopically observed thickness of the biofilm forming on the surface of the foams. Factors that significantly impact the ongoing process are presented using a Pareto chart (Figure 1). The vertical line drawn for the standardized effect value of 2.447 determines the confidence level corresponding to the null hypothesis, i.e., 95% (a = 0.05). Based on the data analysis in Figure 1a, it can be concluded that the biofilm thickness is influenced by factors, such as the inoculation procedure, shaking, and incubation time. However, those below the confidence level, i.e., temperature and pH, turned out to be insignificant. The determination coefficient for the Plackett–Burman analysis is equal to 88.73%. Based on the results obtained, the nature of the correlation between statistically significant variables and the response, i.e., biofilm thickness, was observed. It was found that low values of biofilm thickness, at the level of 40 mm, are formed when the system is intensively shaken at 200 RPM for the longest possible time, 24 h. Moreover, lower values of biofilm thickness were obtained for systems in which inoculation was followed by incubation (Figure 7b). At the same time, negligible effects of pH and temperature (within the investigated ranges) on the research results were observed. However, it is well known that both pH and temperature affect the biotrickling filtration performance to a great extent [77].

Further tests allowed us to observe that the obtained biofilm thicknesses change slightly (within 2%) when the incubation period with shaking increases. Moreover, it was observed that after 10 days of this type of culture, partial death of the biofilm structure occurs, resulting in the appearance of a precipitate, probably containing morphotic remains of microorganisms that are part of the biofilm, and foaming, which may be related to the release of cytoplastic proteins into the suspension. Therefore, day 10 was adopted as the cut-off day for transferring the inoculated foam discs to the target biofilter installation, where, as a result of sprinkling, morphotic remnants are removed on an ongoing basis, preventing the formation of mold outbreaks.

It can be stated that the biofilm structure is already present in the case of BTF A on the first day of biotrickling filtration. After 30 days of biofiltration, further biofilm development was observed, which grew to a greater thickness in BTF B (Figure 8b). Microscopic photos taken under 50× magnification reveal that the thickness of the biofilm layer differs several times. However, the analysis of the biomass dry mass and moisture content allows us to state that a greater increase in biomass occurs in BTF A, i.e., where the biofilm formed a thinner layer. Figure 8c assesses the biofilm condition after 120 days of the process. It can be stated that the biofilm in BTF A adopted a multi-layer structure, and filamentous fungi grow through the neighboring layers, maintaining the entire structure. In the case of BTF B, where the biofilm layer is thicker, no layered systems were observed, and as can be seen from the image of filamentous fungi (Figure 8d), the hyphae grow beyond the biofilm structure, not creating a network with neighboring layers, which favors the formation of the biofilm cracks visible in Figure 8c (BTF B). These results correspond to the dry mass and moisture content in the biofilm structure and to the changes in the surface tension values (Figure 6a). The liquid obtained from BTF A has both lower surface tension and lower zeta potential values obtained during the entire biofiltration process. This can be associated with a greater biomass increase and a thinner biofilm layer formation. In BTF A, the retention of trickling water in the biofilm layer is lower, which means that better removal of metabolic by-products released by microorganisms living in the biofilm layer should be expected. A better exchange of nutrients and removal of proteins, including cytoplasmic proteins, result in lower surface tension of the exhausted liquid. This, in turn, increases the hydrophobicity of the trickling liquid, leading to increased affinity of the liquid phase to gaseous BTEX and, thus, increased absorption capacity. This is revealed by the higher determined headspace concentrations of BTEX in system A than in BTF (Figure 7). Moreover, lower values of the zeta potential for BTF A than BTF B suggest that BTF A operates under less polar/more hydrophobic conditions than BTF B (Figure 6b). According to Qian et al. [27], changing the zeta potential to more positive values corresponds to an increase in the quantity of amino acids and proteins in the biofilm, and easier aggregation of microorganisms in the biofilm is possible, resulting in facilitated growth and development of the biofilm. This aligns with the higher BTEX abatement potential for BTF A than BTF B.

It is recommended that further investigations in this field should cover the following issues: (1) determine which specific species of rot fungi are profitable for the use in biotrickling filters for the enhanced removal of hydrophobic VOCs from air; (2) determine the mass transfer relations for the systems with and without rot fungi addition; (3) testing the procedure proposed in this study (pre-incubation and the use of rot fungi) at bigger scales for real (industrial/municipal) waste gas streams.

## 4. Materials and Methods

Experiments were performed using two similar biotrickling filters (Figure 9), differing in the inoculation procedure. BTF A was packed with polyurethane foam discs and soaked in the microbial-containing medium prior to the introduction of the discs to the biotrickling filter. BTF B was packed with clean polyurethane discs and the packing discs were inoculated by microorganisms via liquid medium circulation through the system. 

Biotrickling filters were made of plexiglass (height of 0.68 m and internal diameter of 0.08 m), packed with polyurethane foam (PU) discs (PPI 10, Murano Feniks, Wejherowo, Poland). The total packing volume was 2.5 dm^3^. VOC vapors were obtained by passing compressed air with a controlled flow rate through liquid layers of VOC mixture (barbotage phenomenon). Such a VOC-rich airstream was mixed with clean air (flow rate was controlled), and the obtained mixture was supplied to the bottom of the biotrickling filter. Mass flow controllers (MFCs) were used to regulate and control the airflow rate, both for generating the VOC vapors and for the clean air (GFC17AVHN6C0 mass flow controllers, Aalborg, Orangeburg, NY, USA). Benzene, toluene, ethylbenzene, and xylene were purchased from Chempur, Poland, and were of analytical grade.

The thermal stability of BTF A and BTF B was set in a range between 20 and 30 °C. Due to the construction of biotrickling filters based on organic glass, the maximum air overpressure supplied was 0.1 bar (g). Previous tests on the fragments of organic glass confirmed that the chemicals and their concentrations used in this study ensured the chemical stability of the system.

Activated sludge (AS) was obtained from a wastewater treatment plant (Choczewo, Pomorskie voivodeship, Poland) and was incubated at 30 °C, 200 RPM, 24 h. The suspension of microorganisms and their remnants were centrifuged (Hettisch, Tuttlingen, Germany; 450 RPM, 4 min) and decanted from the supernatants. Then, the buffers were inoculated with the liquid (9:1 *v*/*v*, respectively) and applied as the trickling liquid for BTF (10% by volume). The obtained liquid was divided into two parts: one was added directly to BTF B; the other was incubated for 24 h with PU discs, which then served as a biofilm carrier in BTF A. 

The procedures for packing preparation, both for selecting the incubation conditions (according to Table 4) and for the biotrickling filtration processes, are schematically shown in Figure 10.

Microorganisms from activated sludge were used as cultures colonizing the biotrickling filters. The liquid circulating in the BTF B system during the inoculation procedure contained mineral salt medium (MSM), microbiological medium (Buffered Peptone Water, Biomaxima, Lublin, Poland) as a source of carbon, nitrogen and phosphorus during the adaptation period, and starter cultures for composting (C). The initial pH was set to 6.8 ± 0.5. White rot fungi extract containing lignin peroxidase enzymes was introduced into the circulating liquid at day 55 from process initiation. It is assumed that the presence of this type of enzyme improves the digestibility of organic compounds, including VOCs.

White rot fungi (WRF) extracts were prepared from lignocellulosic biomass infeed with fungi. Samples of infected biomass with a high decrease in lignin content during the studied period of time were suspended in a medium with pH 3.5 at 30 °C for 24 h. After that period, the medium was centrifuged, and the supernatant containing enzymes from WRF was suspended in the trickling liquid.

Mineral salt medium (MSM) was used as a trickling solution, according to the process conditions given in Table 6. MSM had the following composition: 15.2 g of Na_2_HPO_4_·12H_2_O, 3 g of KH_2_PO_4_, 0.5 g of NaCl, and 1 g of NH_4_Cl, dissolved in 1 dm^3^ of deionized water. All salts were purchased from POCH (Lublin, Poland) and were of analytical grade.

pH of the trickling liquid was determined using a PH-100ATC pH meter (Voltcraft, Hirschau, Germany) equipped with a DJ113 pH electrode (VWR, Darmstadt, Germany). 

The pressure drop across the biofilter packing was monitored daily using a U-tube manometer connected to the inlet gas line through the port located below the packing layer of each BTF system.

Concentrations of benzene, toluene, ethylbenzene, and xylene were determined using a gas chromatograph equipped with a flame ionization detector (Varian CP-3800, Varian Analytical Instruments, Walnut Creek, CA, USA). DB-624 column (60 m × 0.32 mm × 1.80 μm, Agilent Technologies, Santa Clara, CA, USA) was applied. The following parameters of chromatographic analysis were set: oven temperature 100 °C (isothermal conditions), detector temperature 200 °C, carrier gas flow rate 3 cm^3^ min^−1^, split ratio 10. Nitrogen was used as a carrier gas. Concentrations of investigated volatile organic compounds in the headspace of trickling liquid samples were determined using the same method, similar to the procedure described in [78]. 

Values of surface tension of trickling liquid samples were determined using a Krüss K11 tensiometer (Krüss, Hamburg, Germany) via the Wilhelmy method using a rolled PL22 plate. The measurements were taken at 24 ± 2 °C.

The zeta potential of the particles present in the trickling liquid solution (suspension) was determined using the Malvern ZetaSizer Nano ZS (Malvern, AR, USA) system. The DTS 1060 measurement cell was applied. Zeta potential was evaluated using Smoluchowski approximation. Samples of trickling liquid were taken from the BTF systems and diluted 10× with deionized water. Ten measurement runs were performed, and the average value with standard deviation is reported.

CO_2_ measurements were performed using a Grove module (Speed Studio) containing a carbon dioxide sensor (SCD30). This is a precision carbon dioxide sensor based on the Sensirion SCD30 sensor. The measurement range is from 0 ppm to 40,000 ppm, and the measurement accuracy can reach ± (30 ppm + 3%) between 400 ppm and 10,000 ppm. The sensor response time (t90) is 20 s. The sensor uses non-dispersive infrared (NDIR) measurement technology. 

The mass of the dry biomass developed on a packing element, as well as the moisture content in the packing, was determined using the gravimetric method. The moisture was calculated as the difference between the mass of the wet packing element with developed biomass minus the mass of the dry packing element with dry biomass. The mass of dry biomass was obtained by subtracting the mass of dry, fresh, and clean packing elements from the mass of dry elements with developed biomass. For each BTF system, 5 packing elements (polyurethane discs with a diameter of 8 cm and height of 2 cm) were taken from different bed heights (0.2, 0.4, 0.6, 0.8, and 0.95 of the packing height) and subjected to gravimetric evaluation. 

The process efficiency was evaluated based on the removal efficiency (*RE*) and elimination capacity (*EC*) values for given inlet loading of BTEX (IL):*RE = (C_in_ − C_out_)/C_in_,*(2)
*IL = (Q·C_in_)/V*(3)
*EC = RE·IL*(4)
where *RE* is removal efficiency (dimensionless), *IL* is inlet loading (g/(m^3^ h)), *EC* is elimination capacity (g/(m^3^ h)), *C_in_* and *C_out_* are VOCs concentrations in the gas phase at the biofilter inlet and outlet, respectively (ppm), *Q* is the gas flow rate (dm^3^/min), and *V* is the volume of packing (dm^3^).

Empty bed residence time (*EBRT*, min) was calculated by dividing the packing volume by the gas flow rate.

The carbon dioxide production rate (P_CO2_) was described using the following formula:*P_CO2_ = Q·(C_CO2out_ − C_CO2in_)/V*(5)
where *C_CO2out_* and *C_CO2in_* are carbon dioxide concentrations in the biofilter outlet and inlet gas streams.

## 5. Conclusions

The removal of benzene, toluene, ethylbenzene, and xylene (BTEX) from air was investigated in two similar biotrickling filters, differing in the inoculation procedure. Inoculation of the biotrickling filter packing prior to the start of the process (soaking of the pack elements in microbial-rich medium, BTF A) results in a faster process start-up (about 20 days) and higher removal efficiency (up to 96%) than for a biotrickling filter inoculated by a circulating microbial-rich liquid medium (BTF B) (start-up period of 30 days and maximum BTEX removal efficiency of 77%). 

The improved performance of BTF A results in a lower pressure drop (up to 75 and 86 mm H_2_O, respectively) compared to BTF B (75 vs. 86 mm H_2_O, respectively). This is mainly due to developing a thinner biofilm layer and lower water retention in BTF A.

The application of white rot fungi results in a decrease in the surface tension of the trickling liquid (from about 61 to 53 and from 64 to 55 mN/m for BTF A and BTF B, respectively) as well as a decrease in the absolute value of the zeta potential of particles in the liquid (from about −26 to −18 mV and from −27 to −20 mV for BTF A and BTF B, respectively). This indicates more favorable conditions for the removal of hydrophobic VOCs such as BTEX in biotrickling filters supplemented with white rot fungi extracts. 

The novel approach of packing incubation and rot fungi addition investigated for the first time in this work is advisable for modifying existing systems for biological air treatment in industrial and municipal facilities, where BTEX removal is limited due to the mass transfer barrier from the gas to the liquid phase.

## Figures and Tables

**Figure 1 molecules-29-04431-f001:**
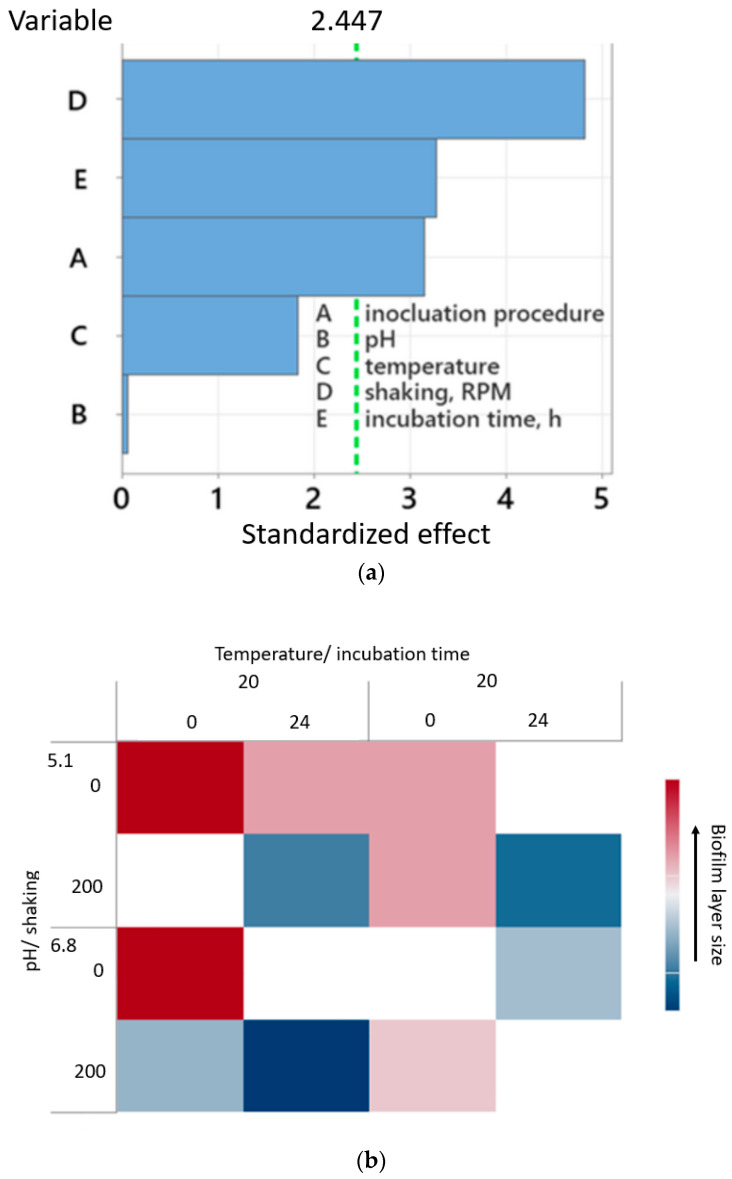
Pareto chart for the standardized effects (biofilm layer as response, a = 0.05) (**a**); correlogram for biofilm layer development concerning binary effects between pH/shaking and temperature/incubation time for experiments with incubation (**b**).

**Figure 2 molecules-29-04431-f002:**
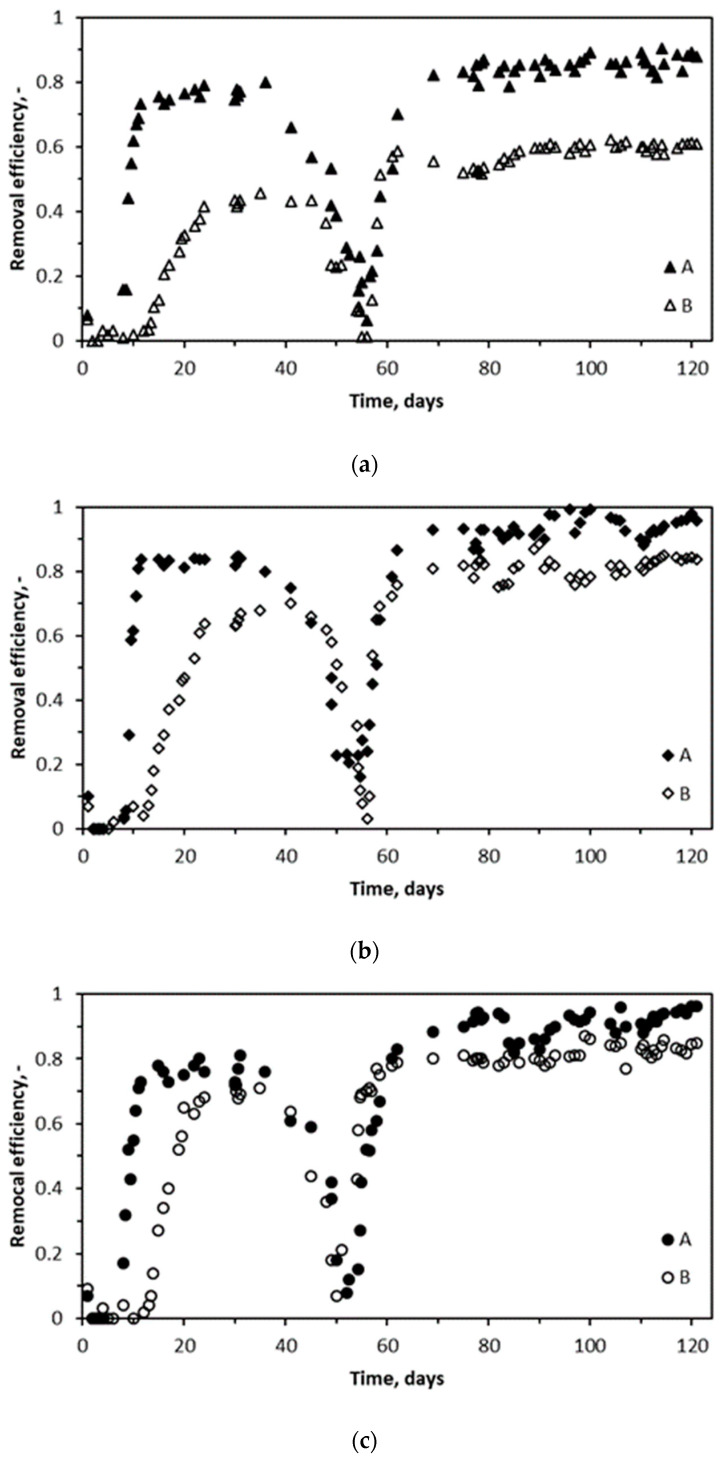
Biotrickling filtration performances of BTF A and BTF B: (**a**) benzene; (**b**) toluene; (**c**) ethylbenzene; (**d**) xylene. Specific process parameters are given in Table 6.

**Figure 3 molecules-29-04431-f003:**
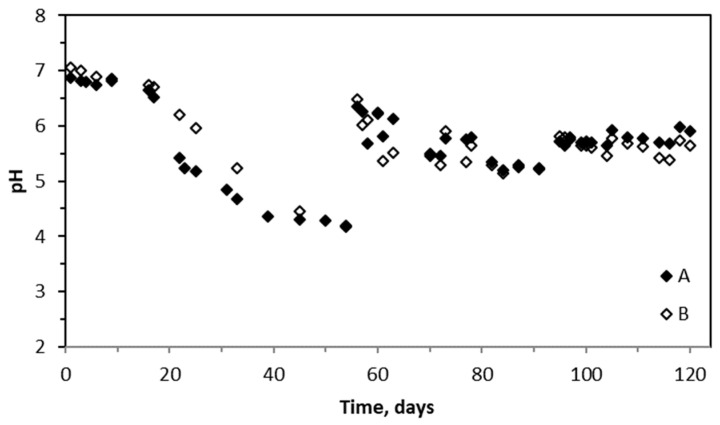
Changes of pH values for BTF A and B.

**Figure 4 molecules-29-04431-f004:**
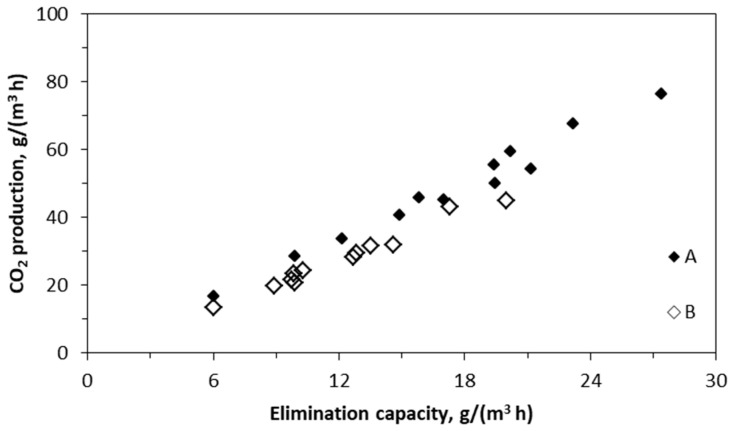
Production of CO_2_ for BTF A and B.

**Figure 5 molecules-29-04431-f005:**
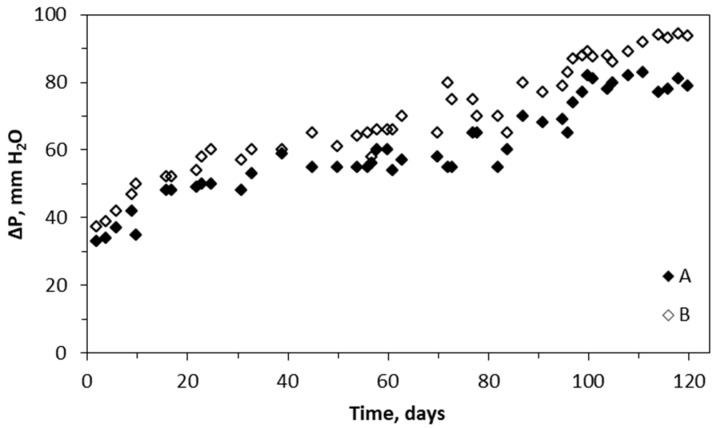
Changes in pressure drop during the biotrickling filtration.

**Figure 6 molecules-29-04431-f006:**
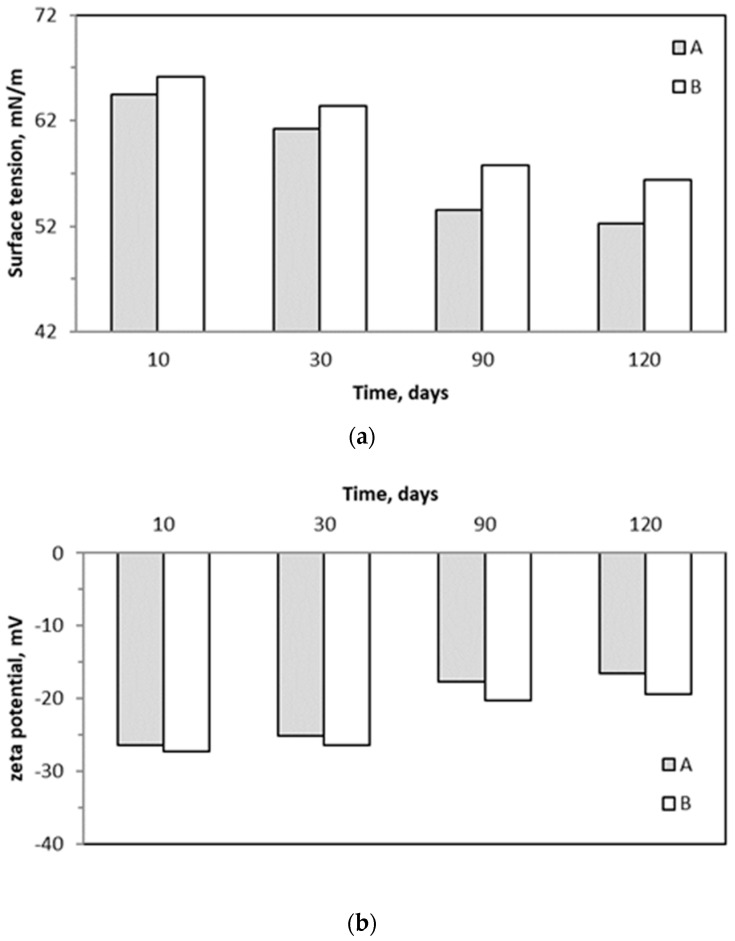
Changes of surface tension (**a**) and zeta potential (**b**) of trickling liquid for BTF A and B.

**Figure 7 molecules-29-04431-f007:**
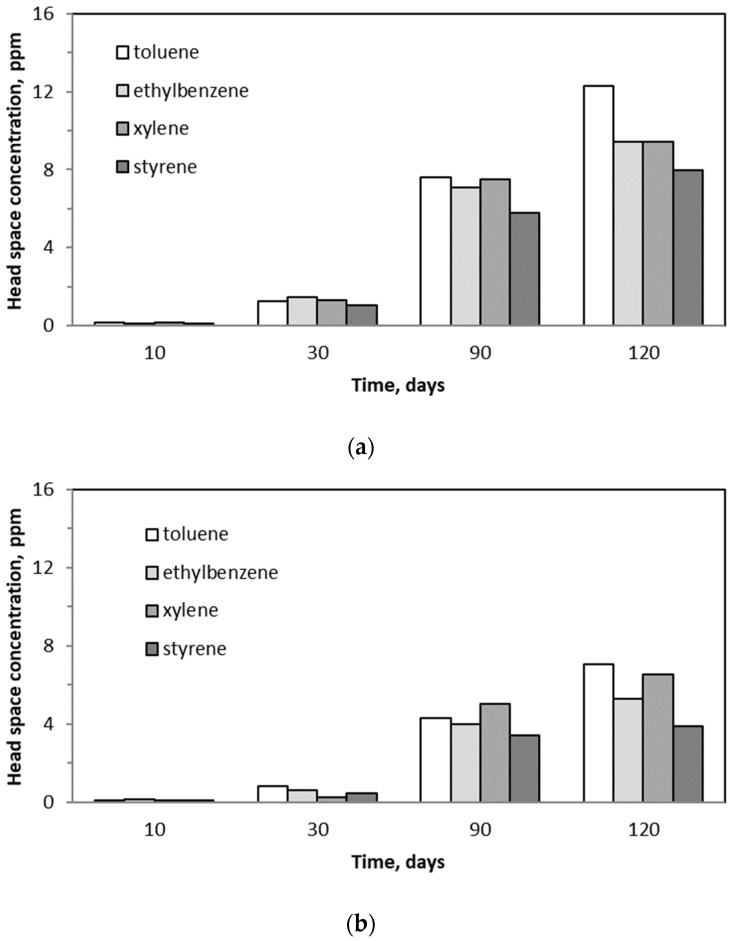
Headspace concentrations of BTEX for trickling liquid samples from BTF A (**a**) and BTF B (**b**).

**Figure 8 molecules-29-04431-f008:**
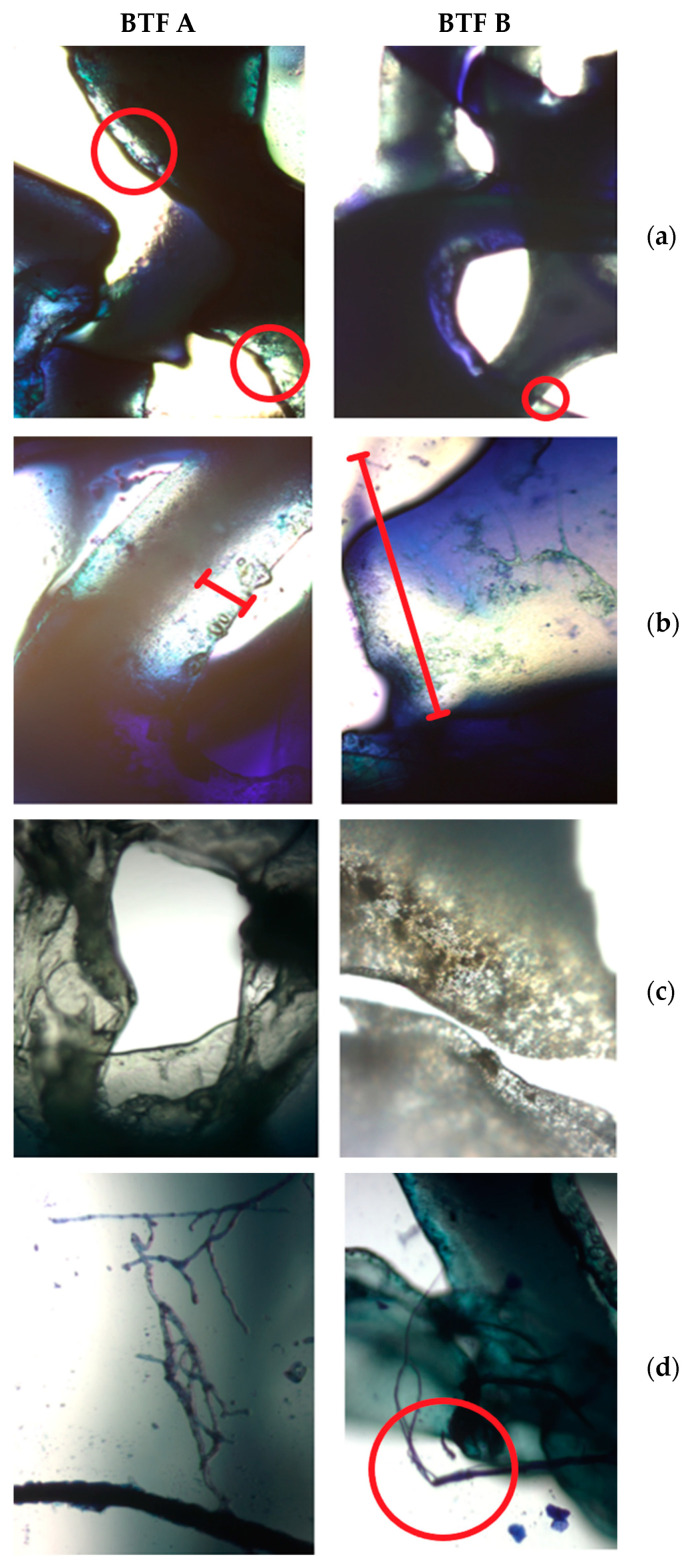
Development of cultures in biofilter for BTF A and B systems: start-up of biotrickling filtration process (**a**); photos after 30 days of biotrickling filtration (**b**) and after 120 days (**c**); final structure of filamentous fungi in biofilm (magnification 50×) (**d**).

**Figure 9 molecules-29-04431-f009:**
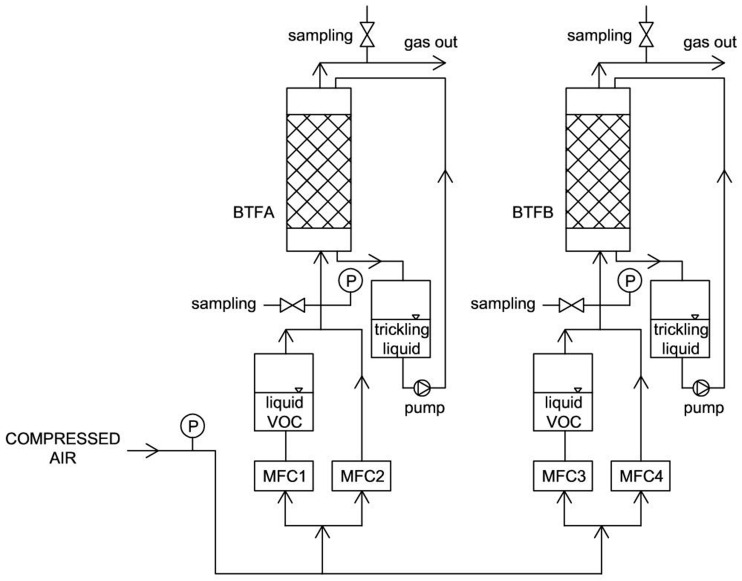
Scheme of the experimental set-up: VOC—volatile organic compound; MFC—mass flow controller.

**Figure 10 molecules-29-04431-f010:**
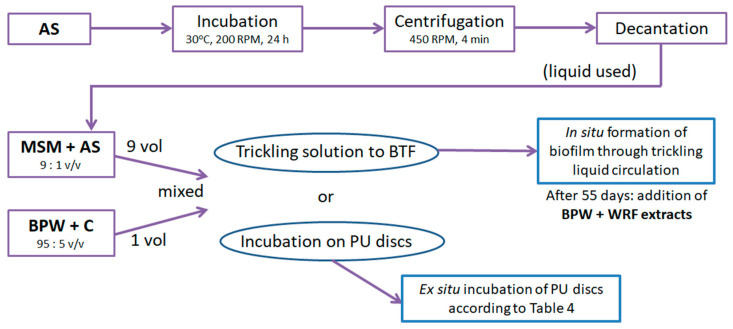
Stages of packing preparation for biotrickling filters: AS—AS-activated sludge; MSM—mineral salt medium; BPW—buffered peptone water; WRF—white rot fungi; C—starting cultures for composting.

**Table 3 molecules-29-04431-t003:** A summary of the lower and upper levels of variables used to assess variability using the Plackett–Burman plan.

No.	Variable	Lower Value (−1)	Upper Value (1)
A	Inoculation procedure	−1 (without incubation)	1 (with incubation)
B	pH	5.1	6.8
C	Temperature, °C	20	30
D	Shaking, RPM	0	200
E	Incubation time, h	0	24

**Table 4 molecules-29-04431-t004:** Plackett–Burman factorial design matrix for examining the influence of five variables.

Run	Inoculation Procedure	pH	Temperature, °C	Shaking, RPM	Incubation Time, h
1.	1	6.8	30	0	24
2.	1	6.8	20	200	0
3.	−1	6.8	30	0	24
4.	1	6.8	20	200	24
5.	1	5.1	20	0	24
6.	−1	6.8	20	0	0
7.	1	5.1	30	0	0
8.	−1	5.1	20	200	24
9.	−1	5.1	20	0	0
10.	−1	5.1	30	200	24
11.	−1	6.8	30	200	0
12.	1	5.1	30	200	0

**Table 5 molecules-29-04431-t005:** Values of variables adopted for biofilter processes according to Plackett–Burmann design.

Biofilter	Inoculation Procedure	pH	Temperature, °C	Shaking, RPM	Incubation Time, h
A	applied	6.8	20	200	24
B	not applied	6.8	20	0	0

**Table 6 molecules-29-04431-t006:** Parameters for biotrickling filtration processes.

Parameter	Value
Packing volume, dm^3^	2.5
Gas flow rate, dm^3^ min^−1^	2.5
Empty bed residence time (EBRT), min	1
Inlet loading (sum of BTEX), g m^−3^ h^−1^: Days 1–40 Days 41–80 Days 81–120	10 ± 0.4 20 ± 0.4 30 ± 0.5
Trickling for BTF A	5.76 dm^3^ h^−1^ (5 s each 10 min)
Trickling for BTF B	Days 1–40: 1.44 dm^3^ h^−1^ (continuous trickling) Days 41–120: 5.76 dm^3^ h^−1^ (5 s each 10 min)
Trickling liquid	Days 1–20: MSM; days 21–55: tap water; days 56–120: MSM
Temperature, °C	22–25 (room temperature)

**Table 7 molecules-29-04431-t007:** CO_2_ production vs. elimination capacity: literature evidence.

Target Compound(s)	(M_CO2/EC_)	Inoculum	Reference
Toluene	1.45	Activated sludge from WWTP	[68]
Toluene	2.84	Aerobic activated sludge	[69]
Toluene	1.99	Fungal consortium (*Trichoderma asperellum* and *Fusarium solani)* with rhamnolipids addition	[70]
Ethanol	1.02	Compost-derived microorganisms with microalgae	[71]
	1.31	Compost-derived microorganisms	
Ethanol	0.44	Compost-derived microorganisms with microalgae on alginate beads	[72]
Ethylbenzene	1.36	Aerobic activated sludge	[69]
Xylene	2.86		[66]
Ethanol, ethyl acetate, MEK	0.30	Activated sewage sludge	[73]
Acetone, toluene, trichloroethylene	0.54	Microbial seeds from WWTP on granular activated carbon	[74]
Mixed VOCs (n-hexane, trichloroethylene, toluene, α-pinene)	0.41	*Candida subhashii*	[75]
Dimehtyl sulfide, propanethiol, toluene	2.26	*Alcaligenes* sp. SY1, *Pseudomonas putida* S1, and microorganisms from activated sludge	[76]
Mixed VOCs (BTEX)	2.37–2.71	Activated sewage sludge, composting starter, and white rot fungi extract	This work

## Data Availability

Specific data not directly reported in the manuscript are available from the authors on request.

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
