# Peer review of "Packing Incubation and Addition of Rot Fungi Extracts Improve BTEX Elimination from Air in Biotrickling Filters"

_molecules, 2024, doi:10.3390/molecules29184431_

Round 1
Reviewer 1 Report
Comments and Suggestions for Authors
In this manuscript entitled “Packing incubation and addition of rot fungi extracts improves performance of biotrickling filter treating air polluted with BTEX” by Rybarczyk et al., the authors systematically compared the effects of inoculation procedure on BTF removal of BTEX. Moreover, they also investigated the effects of using white rot fungi enzyme extracts to enhance the abatement of mixed hydrophobic VOCs in biotrickling filters. The topic of this article is interesting and the results are meaningful, while there are still some issues and concerns in this version which need appropriately addressing prior to a publication in the Molecules. I have listed detailed comments and suggestions for the authors in the follows.
On Abstract
1. The Abstract should be improved with clear quantitative findings and results.
2. Line 19 Page 1: the “VOCs concentrations” should be “VOC concentrations”. There are still many similar errors in the main manuscript (Line 57, 310 450, 451), please verify and revise them carefully.
On Introduction
3. The authors should consider to present the prospective audiences a better and more comprehensive background information on the the reasons for using biofiltration for VOC treatment, the advantages of biofiltration, cases and results of VOC treatment, etc. The following literature might serve these purposes on some aspects: Environmental Science & Technology, 2022, 56: 10349-10360; Chemical Engineering Journal, 2023, 471: 144420; Bioresource Technology, 2024, 398: 130480.
On Results
4. An in-depth analysis should be provided. More comparisons and analyses between the results of this study and those of similar studies should be provided.
5. Figure 2: It is suggested that the authors to provide the reaction conditions, such as residence time, VOC loads, concentrations, temperature, etc.
6. Line 210: “BTA A” should be “BTF A”.
7. Lines 209-213: The descriptions of these sentences are contradictory or unclear. Please verify and carefully revise them.
On Discussion
8. The units of the parameters appearing in equations 1-5 should be provided.
9. Lines 252-255: “In this manuscript, the Authors investigated the possibility of enhancing the......”. Please rewrite the sentences.
10. Lines 397-400: The description of the negligible effect of temperature is inaccurate. Because the authors only set two temperature gradients (20oC and 30oC). It can only be described that the temperature set in this study has little impact on the research.
11. Lines 415 and 427: “biofilter A” should be “BTF A”.
On Materials and methods
12. Lines 474-476: There is no discussion or corresponding results on fungal activity in the article.
13. Lines 443 and 445: Please standardize the abbreviations. The abbreviations of BTF A and BTF B should be provided when they first appear.
Author Response
The detailed responses to the Reviewer's comments are provided in the attached file.

Reviewer 2 Report
Comments and Suggestions for Authors
Dear respected Editor,
The manuscript entitled: “Packing incubation and addition of rot fungi extracts improves the performance of biotrickling filter treating air polluted with BTEX” reported the air purification from BTEX using two different post-treated activated sludge-modified polyurethane-packed biotrickling filters. After reviewing this manuscript, it needs some modifications to become fit for publication.
The manuscript will be reviewed again after major revision. Specific comments are listed below:
1- I recommend changing the title to:
“Design high-performance biotrickling filter for air purification”
2-Both the abstract and conclusion need to be rewritten concisely and quantitively.
3- Please revise the English writing throughout the whole manuscript. Please double-check on grammar, missing articles, punctuation, and so on.
4-Please mention the novelty of this paper in detail by the end of the introduction comparing it with other reported studies
5-The quality of inserted figures needs to be improved
6-Design the schematic preparation diagram including real sample images
7-Regeneration of the designed filters is missing
8-Thermal, chemical, and mechanical stabilities of the prepared filters are missing

Please revise the English writing throughout the whole manuscript. Please double-check on grammar, missing articles, punctuation, and so on
Author Response
The detailed responses to the Reviewer's comments are given in the attached file.

Round 2
Reviewer 2 Report
Comments and Suggestions for Authors
No